# Macrophages in the Context of Muscle Regeneration and Duchenne Muscular Dystrophy

**DOI:** 10.3390/ijms251910393

**Published:** 2024-09-27

**Authors:** Francisco Hernandez-Torres, Lidia Matias-Valiente, Virginia Alzas-Gomez, Amelia Eva Aranega

**Affiliations:** 1Department of Biochemistry and Molecular Biology III and Immunology, Faculty of Medicine, University of Granada, 18016 Granada, Spain; fhtorres@ugr.es; 2Medina Foundation, Technology Park of Health Sciences, 18016 Granada, Spain; lmmatias@ujaen.es (L.M.-V.); virginiaalzas@gmail.com (V.A.-G.); 3Department of Experimental Biology, Faculty of Experimental Sciences, University of Jaen, 23071 Jaen, Spain

**Keywords:** macrophages, inflammation, skeletal muscle, Duchenne muscular dystrophy

## Abstract

Macrophages are essential to muscle regeneration, as they regulate inflammation, carry out phagocytosis, and facilitate tissue repair. These cells exhibit phenotypic switching from pro-inflammatory (M1) to anti-inflammatory (M2) states during muscle repair, influencing myoblast proliferation, differentiation, and myofiber formation. In Duchenne Muscular Dystrophy (DMD), asynchronous muscle injuries disrupt the normal temporal stages of regeneration, leading to fibrosis and failed regeneration. Altered macrophage activity is associated with DMD progression and physiopathology. Gaining insight into the intricate relationship between macrophages and muscle cells is crucial for creating effective therapies aimed at treating this muscle disorder. This review explores the dynamic functions of macrophages in muscle regeneration and their implications in DMD.

## 1. Introduction

Macrophages are the most adaptable cells within the hematopoietic system, serving as the body’s first line of immune defence against pathogens. Classically, they have been recognized as key components of the innate immune system, primarily responsible for triggering and resolving tissue inflammation. For many years, they were considered a homogeneous group of phagocytes, sharing a common origin and performing similar roles. Nonetheless, this concept has been renewed in recent decades. It is now understood that these versatile cells play crucial roles in various biological processes, including tissue remodelling during organ development, maintaining tissue homeostasis, repairing damaged tissues, clearing dead cells, promoting wound healing, and mounting immune responses to pathogens [1,2,3,4,5,6,7]. In this review, we will examine the role of macrophages in maintaining and supporting healthy muscle function, highlighting the presence of distinct macrophage subpopulations in this tissue. Furthermore, we will examine the specific functions of M1-like and M2-like macrophages in immune responses and tissue repair, shedding light on the dynamic nature of macrophage polarization in vivo. We will discuss how this polarization is altered in the dystrophic muscle and how this contributes to the worsening of the disease. Finally, we will discuss the potential of macrophage behaviour modulation as a therapeutic tool in DMD. In summary, this review aims to serve as a resource of macrophage biology in muscle homeostasis as well as to explore their role in DMD progression.

## 2. Macrophages in the Skeletal Muscle: Different Origins and Subtypes

Since macrophages were first characterized by Elie Metchnikoff in 1893, their origin has generated a great deal of controversy [8]. In 1972, van Furth et al. proposed for the first time the “mononuclear phagocyte system” theory, which emphasized that all macrophages, either resident or infiltrating bone marrow (monocyte)-derived macrophages (BMDMs), are derived from blood monocytes [9]. Nonetheless, today, we know that this is partly inaccurate. The role of these cells, both in muscle homeostasis and its role on certain pathologies, has also been debated and reviewed in recent decades. In this regard, the phenotype that these cells present has emerged as a critical issue determining their role.

In the late 20th century, Mills and colleagues introduced the concept of macrophage polarization, known as the M1/M2 classification system, which was based on macrophage responses to different stimuli in vitro [10,11]. This concept suggests that macrophage plasticity can be generally divided into two phenotypes: M1 (classically activated) and M2 (alternatively activated) [12]. In the context of tissue repair (including muscles), a very basic first classification included M1 macrophages as having a pro-inflammatory phenotype capable of damaging host tissue, whereas M2 were considered anti-inflammatory and pro-regenerative macrophages [13,14]. This nomenclature was mainly based on macrophage induction by specific mediators in vitro but is less applicable to macrophages in an in vivo context where stimuli/cytokines that promote either M1 or M2 phenotypes often coexist. Indeed, evidence suggests that macrophages can exhibit a combination of M1 and M2 phenotypes and may not necessarily expand clonally to sustain this phenotype [15,16,17]. Therefore, the phenotype of in vivo macrophages could be considered M1-like or M2-like but not exclusively M1 or M2 [18]. In this section, we will discuss the latest scientific advances that help us to understand the origin and functions played by macrophages in muscle at the steady state.

### 2.1. Skeletal Muscle-Resident Macrophages

Until recently, it was widely accepted that macrophages followed a consistent colonization pattern across most tissues during both embryonic and adult stages (Figure 1).

However, recent findings have made this paradigm a little more complicated. Recently, Wang et al., have identified cell populations of skeletal muscle-resident macrophages in mice by using single-cell RNAseq analysis [26]. These authors defined as muscle-resident macrophages a subpopulation of CD45^+^/CD11b^+^/F4/80^+^/CD64^+^/LY6C^lo^/MerTK^+^/CD11C^−^/CD163^+^/CD206^+^ cells that are found in the interstitial tissues, such as the epimysium, perimysium, and endomysium [26]. Using cell lineage tracing and bone marrow (BM) transplant experiments, Wang et al. provided strong evidence that these skeletal muscle-resident macrophages arise from different sources: (a) yolk sac (YS) haematopoiesis, derived from primitive macrophages whose origins are early erythro-myeloid progenitors (EMPs) that emerge in the YS during embryonic stages (Figure 1A); (b) foetal liver haematopoiesis, derived from monocytes whose origins are hematopoietic and non-hematopoietic stem cells at foetal stages (Figure 1B); and (c) postnatal BM haematopoiesis, derived from adult monocytes whose origins are hematopoietic stem cells at adult stages (Figure 1C). Wang et al. propose that resident macrophages in skeletal muscles have similar functions but exhibit different properties from those observed for resident macrophages in other tissues [26]. Like in many other tissues, these authors identified functionally distinct subsets of macrophages within skeletal muscle, each correlating with their origins. Thus, Ccr2^+^/MhcII^hi^/Lyve1^lo^ macrophage clusters primarily originate from adult blood monocytes. On the other hand, Ccr2^−^/MhcII^lo^/Lyve1^hi^ macrophage clusters, which in turn are made up of different subclusters (“Cluster 0”, “Proliferating cluster”, “Cd209 cluster”, and Klf2 cluster), are derived from both embryonic and adult progenitors. Both subsets appear to play roles in muscle repair and homeostasis.

However, functional enrichment analysis revealed that genes differentially expressed by the Ccr2^+^ cluster were notably associated with antigen processing and presentation pathways. This cluster also showed relatively lower expression of “M2-like” genes compared to other non-Ccr2 clusters. In contrast, genes differentially expressed by the non-Ccr2 clusters were enriched in pathways related to phagocytosis and metabolism. Interestingly, the non-Ccr2 Cd209 subcluster exhibited high expression of genes typically upregulated in alternatively activated macrophages (M2) [26].

More recently, another single cell-RNAseq analysis performed by Krasniewski et al., also based on the expression of *MhcII* and *Lyve1* genes, divided CD45^+^/CD11B^+^/F4/80^+^ skeletal muscle macrophages into four subgroups with strong phagocytic capabilities: the well-known M1-like (Lyve1^−^/MhcII^hi^) and M2-like (Lyve1^+^/MhcII^lo^) macrophages and two additional new subgroups (Lyve1^+^/MhcII^hi^ and Lyve1^−^/MhcII^lo^) that were confirmed by flow cytometry and immunohistology [27]. The newly identified Lyve1^+^/MhcII^hi^ subgroup demonstrated gene expression patterns and functional characteristics resembling both M1 and M2 macrophages. In contrast, the new Lyve1^−^/MhcII^lo^ subpopulation was predicted to present a more pronounced “killing” capability and could play a direct role in innate immunity [27]. In addition, the authors also showed that messenger ribonucleic acid (mRNA) encoding proteins involved in the chemotaxis of granulocytes and monocytes and the cellular response to IFN-γ and M2-like markers were significantly lower in skeletal muscle macrophages from older mice, while mRNA encoding proteins related to cellular detoxification, inflammation, senescence, and long-chain fatty acid transporters were elevated, revealing a dynamic polarization of functional subpopulations of macrophages, highlighting shifts towards pro-inflammatory and senescent phenotypes during ageing [27].

In parallel, over time, Babaeijandaghi et al. outlined the diversity, turnover, and origin of adult muscle-resident mononuclear myelomonocytic (MRMM) populations under normal conditions [28]. The authors discovered that *Timd4* and *Lyve1* were two surface markers highly expressed by these cells under normal conditions yet nearly absent from infiltrating cells, thus indicating that *Timd4*, either alone or in combination with *Lyve1*, could be used to identify resident macrophages in muscle tissue. Additional flow cytometric analysis with the TIMD4 marker identified three primary subpopulations of mononuclear myelomonocytic cells present in intact skeletal muscle: F4/80^+^/TIMD4^+^/LYVE1^+^ (TIMD4+) macrophages, F4/80^+^/TIMD4^−^/CX3CR1^+^ (TIMD4^−^) macrophages, and a population of F4/80^Low^/CD11C^+^/MHCII^+^ (CD11C+) cells, likely representing dendritic cells, being TIMD4^−^ and TIMD4^+^ populations the majority of MRMM cells. Using parabiosis experiments and lineage-tracing approaches, the authors showed that F4/80^+^/TIMD4^+^/LYVE1^+^ cells constitute a population of locally self-renewing resident macrophages (SRRMs), whereas the two other populations, F4/80^+^/TIMD4^−^/CX3CR1^+^ resident macrophages and F4/80^Low^/CD11C^+^/MHCII^+^ dendritic cells, are replenished by circulating blood cells. The cluster F4/80^+^/TIMD4^+^/LYVE1^+^ exhibited high expression levels of genes associated with endocytosis, angiogenesis and vascular remodelling, cell migration, chemotaxis of inflammatory cells, and responses to IL-1/TNF-A signalling. The authors stated that the absence of F4/80^+^/TIMD4^+^/LYVE1^+^ impairs muscle regeneration, because they are essential for the effective removal of apoptotic cells during muscle repair. Additionally, they showed that F4/80^+^/TIMD4^−^/CX3CR1^+^ cells were enriched in antigen-presenting and response to peptidoglycan pathways, lipopolysaccharide-mediated signalling, and the inflammatory response to antigenic stimuli. This suggests that this particular group of muscle-resident macrophages plays a central role in the innate immune response to pathogens [28]. Although those transcriptomic data provide a deeper characterization of different macrophage subpopulations originated from different cell sources during embryonic, foetal and adult stages, additional experiments are necessary to better understand how different origins determine their functions.

### 2.2. Infiltrating Macrophages

Following muscle injury, substantial necrosis of muscle fibers is evident on the first day, accompanied by the infiltration of inflammatory cells, such as macrophages. Inflammation reaches its peak on day three. On day five, small, centrally nucleated myoblasts and multinucleated myotubes begin to appear, and by day seven, the number of infiltrating inflammatory cells decreases substantially, and necrotic fibers are largely replaced by regenerating fibers. At this stage, transient fibrosis develops, highlighted by an increase in extracellular matrix (ECM) deposition, aimed to provide structural support for injury repair. Between days 14 and 21, both inflammation and temporary fibrosis subside, and the muscle fibers return to a size similar to that of uninjured muscle. A proper inflammatory response, driven primarily by macrophage infiltration, is crucial for the repair of acute skeletal muscle injuries. In fact, it has been extensively shown that without macrophage infiltration or if macrophage functions are disrupted, muscle regeneration is severely compromised, leading to significant muscle fibrosis [14,29,30,31,32,33,34].

It has been shown that after muscle injury, the phagocytosis of damaged muscle fibers is not mediated by resident macrophages [35]. Instead, they have been proposed to serve as sentinels, activated by damage-associated molecular patterns (DAMPs) during injury, to aid in the recruitment of circulating leukocytes [35,36,37,38] (Figure 2). The infiltration of BMDMs can conduct phagocytosis, contributes to the accumulation of intramuscular LY6C^−^ macrophages, and produces a high level of IGF-I. Overall, these findings suggest that resident macrophages fall short in fully performing pro-regenerative functions, highlighting the essential role of infiltrating BMDMs in skeletal muscle regeneration following acute injury [32].

As in many other tissues, in the injured muscles, the recruitment of LY6C^hi^ inflammatory monocytes requires the expression of the CCR2 receptor on the surface of these monocytes (Figure 2). In addition, the expression of CCL2, the main ligand of CCR2, is required in both muscle-resident cells and infiltrating macrophages to reach muscle tissue [30,31,32].

It has been shown that LY6C^hi^ inflammatory macrophages arrive to injured muscle at day one post injury and, after phagocytosing necrotic muscle debris, they shift to LY6C^lo^ macrophages, becoming the main macrophage population from the third day of regeneration onwards [14,36,39] (Figure 2). We have mentioned before that M1-like or M2-like phenotype classification is not clearly established in vivo. Therefore, it has been demonstrated that in injured skeletal muscle, LY6C^hi^ macrophages in the early stages of inflammation do not fit the strict M1 classification, while LY6C^lo^ macrophages in later stages do not exclusively represent M2 macrophages [40,41,42]. Regarding this, it has been demonstrated that, although the transition from LY6C^hi^ to LY6C^lo^ macrophages involved the downregulation of M1 genes (tnfa, il1b, and il6) and the upregulation of M2 genes (cd206, tgfb1, and igf1), LY6C^hi^ macrophages on day one co-expressed high levels of both M1 (tnfa, il1b, and il6) and M2 genes (arginase 1, ym1, and il10) [42]. 

Varga et al. suggested the existence of four main features of muscle-derived macrophages that play crucial roles in regeneration: (1) infiltrating LY6C^+^ macrophages, which express acute-phase proteins and display an inflammatory profile independent of IFN-γ, function as damage-associated macrophages; (2) macrophage metabolic changes, including reduced glycolysis and increased activity in the tricarboxylic acid cycle/oxidative pathway, precede and support the transition to an anti-inflammatory profile; (3) LY6C^−^ macrophages, derived from LY6C^+^ cells, show active proliferation; and (4) later, restorative LY6C^−^ macrophages exhibit a distinct profile, characterized by the secretion of molecules involved in intercellular communication, particularly matrix-related molecules [41]. Taken together, these findings underscore the dynamic nature of macrophage responses at the molecular level following acute tissue injury and the subsequent repair process.

## 3. Roles of Macrophages on Skeletal Muscle Homeostasis and Muscle Repair

### 3.1. Macrophages Contribute to Initiate but Also Resolve Inflammation

As indicated above, during muscle regeneration, macrophages participate as effectors and regulators of the inflammatory response, playing a crucial role in regulating both the onset and resolution of inflammation during the repair process of acute skeletal muscle injury. During the initial stages after muscle injury, LY6C^hi^ monocytes/macrophages rapidly infiltrate the damaged tissue, producing significant levels of pro-inflammatory cytokines, such TNF-α and IL-1β (Figure 2). These cytokines promote inflammation, amplifying inflammatory cell recruitment and thus increasing tissue damage [14,42,43,44]. Pro-inflammatory macrophages also play an active role in phagocytizing and clearing damaged tissue debris for muscle injury repair [14,42,43,44,45] (Figure 2). It is important to note that different experiments, in vitro and in vivo, have shown that several molecular effectors, such as MKP1, AMPK, METRNL, IGF-1, and transcription factors CEBPB and NFIX, and the phagocytosis of muscle cell debris after muscle damage promote a switch of pro-inflammatory toward an anti-inflammatory macrophage phenotype [14,46,47,48,49,50,51,52] (Figure 2). Subsequently, anti-inflammatory macrophages contribute to inflammation resolution by releasing anti-inflammatory cytokines, such as IL-4, TGF-β1, and IGF-1 [36,42] (Figure 2). These anti-inflammatory molecules reduce pro-inflammatory signals, lower reactive oxygen species production, prevent neutrophil recruitment, and promote neutrophil apoptosis and its subsequent removal by macrophages [36,53]. In addition to these infiltrating LY6C^hi^ monocytes/macrophages, Babaeijandaghi et al. have been recently shown that a new phenotype, F4/80^+^LYVE1^+^TIMD4^+^ SRRM, is also essential for clearing apoptotic cells that result from damage after an acute injury. In their absence, necrotic fibers are likely to build up and remain throughout the regeneration process [28].

In this section, it is also interesting to highlight the role that biologically active scaffolds can exert in balancing pro-inflammatory and anti-inflammatory macrophage populations in muscular contexts. In this sense, Borrego et al. recently showed in vitro how the secretomes of fibrin-primed bone marrow cells (F-BMCs) induce a switch from a pro-inflammatory to an anti-inflammatory phenotype in macrophages. In addition, they showed that the secretomes of these anti-inflammatory F-BMC-educated macrophages induce cardiomyoblast proliferation and spreading [54]. The authors also stated that this mechanism explains the improvement in cardiac function and the decrease in infarct area in rat hearts when combined treatment of fibrin with unfractionated BM cells is administered sub-chronically in a rat model of myocardial infarction in vivo [54]. Although this phenomenon has been described in cardiac muscle, it would be interesting to analyse the effect that this induction may have in other muscular contexts, since the confirmation of this fact could help to generate new tools aimed at minimizing inflammatory processes related to skeletal muscle function and pathology.

### 3.2. Macrophages Participate in Muscle Stem Cell Activation and Differentiation

Muscle stem cells (MuSCs), also termed satellite cells, are chiefly responsible for the upkeep and repair of muscle tissue [55,56,57,58]. These MuSCs are located underneath the basal lamina and adjacent to the plasma membrane of the skeletal muscle myofiber [59]. In healthy muscle tissue, most MuSCs are in a quiescent state, marked by the presence of the transcription factor PAX7 [60]. In response to physiological stimuli like physical exercise or pathological conditions, MuSCs become activated, leading to their proliferation, differentiation, and fusion into multinucleated myofibers, thereby facilitating effective regeneration [55,56,57,58] (Figure 2). Following muscle injury, the damaged tissue releases several molecular signals that activate MuSCs, enabling the translation of *Myf5* and *Myod1* mRNAs [55,61]. Soon after, activated MuSCs divide asymmetrically to produce both Pax7^+^/MYF5^+^ and Pax7^+^/MYF5^−^ cells [62]. While Pax7^+^/Myf5^−^; cells contributed to the maintenance of the MuSC pool (self-renew), activated Pax7^+^MYF5^+^MYOD^+^ cells proliferate and differentiate by downregulating PAX7 and MYF5 and upregulating MYOG and MYF6 proteins, thus leading to the formation of myocytes that ultimately fuse each to other to generate myofibers [55,63,64,65].

Macrophages also play important roles in this process. This affirmation is supported by much in vitro and in vivo evidence (Figure 2). Thus, in vitro co-culture studies have demonstrated that direct physical interaction between macrophages and MuSCs inhibits apoptosis in myogenic cells [66,67]. More recently, Juhas et al. demonstrated that the incorporation of macrophages into muscle tissues engineered from adult-rat myogenic cells allows for nearly complete structural and functional repair following cardiotoxic injury in vitro [68]. In this article, the authors clearly demonstrated that, when macrophages are absent, engineered muscle derived from adults does not effectively repair itself after injury, even when they are treated with pro-regenerative cytokines. In contrast, the presence of BMDMs within the in vitro engineered muscles stimulates MuSC-mediated myogenesis by significantly limiting myofiber apoptosis and degeneration and increasing blood vessel ingrowth, cell survival, muscle regeneration, and contractile function. The authors state that this effect is at least in part contributed by macrophage downregulation of paracrine pro-inflammatory cytokines such TNFα and IL-1β, known to be associated with muscle wasting and degeneration [68] (Figure 2).

It has been demonstrated in vivo that pro-inflammatory (M1-like) macrophages stimulate myoblast proliferation but hinder their fusion and differentiation, whereas anti-inflammatory (M2-like) macrophages suppress myoblast proliferation while encouraging myotube formation and differentiation. These effects are partly driven by paracrine cytokines and growth factors secreted by macrophages (Figure 2). Thus, it has been shown that pro-inflammatory macrophages can stimulate MuSC proliferation and activation during the early stages of muscle regeneration by releasing several paracrine signals, such as ADAMTS1, PGE2, TNF-α, G-CSF, and IL-6 [69,70,71,72,73] (Figure 2). ADAMTS1 metalloproteinase activity reduces Notch signalling, inducing MuSC activation [72]. PGE2 acts directly on MuSCs through the EP4 receptor, promoting the expansion and engraftment of these cells [71]. TNF-α acts as a mitogen in skeletal muscle, inducing MuSC proliferation and activation by inducing the ERK kinase pathway and activating SRF, respectively, which stimulates the expression of the early response gene c-fos [69,73]. Finally, IL-6 and G-CSF stimulate myoblast proliferation by activating the STAT3 pathway [70,74]. IL-6 also activates the STAT3 pathway in macrophages in an autocrine manner. This activation induces in macrophages the production of chemokines CCL2 and CCL3, thus stimulating further macrophage infiltration into injured muscle through their interaction with CCR2 and CX3CR1 receptors present in blood monocytes, respectively [70] (Figure 2). Conversely, molecules that are prominently expressed by anti-inflammatory macrophages on day three, such as IL-4, GDF-3, IGF-1, and OSM, stimulate myoblast differentiation and myofiber growth [32,75,76,77] (Figure 2). IL-4 acts as a myoblast recruitment factor that increases the number of nuclei and the size of new forming myotubes [75]. GDF3 is a secreted ligand of the TGF-β superfamily that facilitates the restoration of skeletal muscle integrity by encouraging the fusion of muscle progenitor cells [78]. IGF-1 plays an important role in recovery from muscle atrophy and improves muscle regeneration by activating muscle progenitors to help myogenesis [32,50,51,76]. Finally, it has been shown that myogenic precursor cells respond to OSM through both gp130/LIFR and gp130/OSMR receptors, thus stimulating all steps of myogenesis in vitro [77].

### 3.3. Macrophages Conditionate Muscle Repair-Microenvironment

Muscle microenvironment significantly influences muscle fiber growth and myogenic differentiation of MuSCs. ECM in the muscle is of great significance to tissue behaviour, playing an important role in maintaining muscle homeostasis and regulating the developmental myogenesis [79,80]. The main source of ECM components in the muscle are the fibro/adipogenic progenitors (FAPs), a muscle interstitial mesenchymal cell population which regulates MuSC differentiation during tissue regeneration, clearance of necrotic debris, and mediation of ECM deposition [81,82,83]. In fact, disrupted regenerative processes and/or chronic damage may cause abnormal FAP activation, leading to increased fibro-fatty tissue formation and decreasing MuSC activation [84]. The critical role of infiltrating macrophages in modulating FAP activity and ECM remodelling is supported by several experimental data that show how macrophage depletion, or inhibition of macrophage recruitment, not only impairs muscle regeneration but also promotes muscle fibrosis [31,85]. Pro-inflammatory macrophages regulate FAP accumulation and activation after acute skeletal muscle injury by DMD secreting TNF-α and inducing FAP apoptosis to limit excessive FAP accumulation during the initial stages of muscle regeneration [85] (Figure 2).

Wang et al. discovered and detailed the presence of CD45^+^/collagen I^+^ fibrocytes in skeletal muscle during the later stages of acute injury repair. These cells constitute a subset of macrophages, mostly LY6C^lo^ that expressed F4/80 and CCR2, and are most likely recruited from circulation and differentiate when they reach the injured muscles (Figure 2). They expressed a low level of collagen genes (*Col1a*, *Col3a*, and *Col6a*) and a high level of pro-fibrotic growth factor genes, including *Pdgfα*, *Tgf-β1*, and *Pdgfβ*, thus contributing to the necessary transient fibrosis and reparative ECM remodelling during muscle injury repair [86] (Figure 2). The intramuscular fibrocytes do not overlap with FAPs, as they do not express CD34 or PDGFRα. It has been suggested that they could boost the repair functions of muscle fibroblasts, given that TGF-β1 and PDGF are powerful growth factors that stimulate fibroblast activation and activity (Figure 2). This might in turn contribute to a temporary rise in ECM production and deposition, which aids in the tissue repair process (X. Wang et al., 2016). Collectively, all these data highlight the potential of different macrophage subtypes regulating muscle stem cells and/or FAP function. The development of new loss and gain of function in vivo models for those specific macrophage subpopulations could help to further characterize their role in muscle repair.

## 4. Macrophages in DMD Microenvironment

DMD is a severe, progressive disorder that impacts around one in five thousand live male births [87,88,89,90]. This condition arises from mutations in the dystrophin gene located on the X chromosome, leading to muscle degeneration. The lack or malfunction of dystrophin induces sarcolemma instability and myofiber degeneration because myofibers become fragile and easily damaged during muscle contraction. Muscle degeneration leads to chronic inflammation and replacement of muscle tissue with fibroadipose tissue [91,92]. The dystrophic phenotype is further exacerbated by impaired muscle regeneration, since dystrophin is essential for the asymmetric division of MuSCs and the generation of committed myogenic progenitors [93,94,95]. In addition, a significant contribution of muscle injury in the dystrophin-deficient phenotype is also due to secondary damage driven by the immune response [96,97,98,99].

As we mentioned above, in vivo research has demonstrated that macrophage phenotype switch occurs in skeletal muscle after acute injury. Initially, macrophages exhibit a phagocyte-M1-like phenotype, but from day three onwards, they turn into an M2-like phenotype that participates in muscle growth, differentiation, and regeneration [13,14]. However, subsequent data showed that this scenario is markedly different in DMD, in which multiple muscle injuries occur asynchronously in the absence of effective regeneration. This scenario induces a chronic inflammatory state in the muscle, characterized by a fibrotic signature [100]. In this scenario, Villalta et al. proposed that the shifts in macrophage phenotypes in muscular dystrophy are more complex [45]. Thus, in vivo and in vitro experiments performed by Villalta et al. in an *mdx* mouse model indicate that, concomitant with the acute peak of muscle inflammation in mdx mice at 4 weeks, a marked increase of M1 macrophages (iNOS^high^/IFN-γR^+^ phenotype) and M2a macrophages (Arg^high^/IL-4R^+^/CD206^+^ phenotype) is observed in the dystrophic muscle [45]. Subsequently, as DMD pathology progresses (8 weeks, mdx mice), these authors stated that elevated levels of IL-4 and IL-10 lead to the deactivation of M1 macrophages and a decrease in inflammation-associated muscle membrane damage. Elevated IL-10 production may subsequently shift macrophages towards an M2c phenotype (iNOS^low^/Arg^low^/CD163^+^/CD206^+^), a state linked to tissue repair and remodelling through the increased release of anti-inflammatory cytokines such as IL-10, IL-1, and IL-1Ra [45]. However, at the later stages of pathology, another shift toward the M2a phenotype occurs, maintaining high arginase expression. This high arginase activity can exacerbate fibrosis in the mdx muscle by promoting ornithine production, which in turn boosts collagen deposition [45,101,102,103,104,105].

Nonetheless, other findings collectively suggest that a permanent pro-inflammatory macrophage phenotype could contribute to irreversible fibrosis in the later stages of the disease. Thus, in the mdx mouse model for DMD, several experimental studies have shown that macrophages are linked to the development of muscle fibrosis. Fibrinogen, a soluble acute phase protein released into the bloodstream during stress, not only plays a critical role in blood clotting after vascular injury but also extravasates at sites of inflammation [106]. In this regard, Vidal et al. have shown how fibrinogen-Mac-1 receptor binding, through the induction of IL-1β, stimulates TGF-β production by mdx macrophages, which subsequently promotes collagen synthesis by mdx fibroblasts. Fibrinogen-induced TGFβ enhances collagen production by activating profibrotic macrophages. In addition, fibrinogen directly stimulates collagen production by binding to its αvβ3 receptor in fibroblasts. Collectively, these findings reveal the influence of profibrotic macrophages on fibrinogen deposition in muscle dystrophy [107]. In agreement with these findings, Juban et al. have shown that DMD-derived LY6C^+^ macrophages display profibrotic properties by promoting sustained collagen I production in fibroblasts [108]. These authors state that macrophages sustain latent TGF-β1 production due to the higher expression of LTBP4, and *LTBP4* polymorphisms are associated with the progression of fibrosis in patients with DMD. The exacerbation of macrophage-mediated pro-inflammatory stimuli in mdx mice has also been refuted by Acharyya et al., who have shown that the NF-κB pathway is activated by mdx macrophages to promote inflammation and muscle necrosis [109]. Additional studies also support the crucial role of macrophages in maintaining chronic inflammation in the dystrophic context, showing that the prevention of the entry of circulating monocytes into the dystrophic muscle by antibody depletions of mdx macrophages, CCR2 deficiency, or TLR4 ablation temporarily improves muscle histology and function [110,111,112].

More recent findings have revealed that, although it has been described that inflammation in mdx muscles is predominantly driven by macrophage infiltration, recruited from blood LY6C^hi^ monocytes that subsequently turn into LY6C^lo^ macrophages [110,113], the use of simple markers such as LY6C to assess macrophage status in mdx mice may be less reliable.

In fact, gene expression analyses of LY6C^+^ and LY6C^–^ macrophages from both non-fibrotic and fibrotic dystrophic muscles (mdx and *sgca*(−/−) mice) have shown that the so-called pro-regenerative LY6C^–^ macrophages also exhibit elevated levels of pro-inflammatory markers. This suggests that dystrophic muscle contains a mixed-macrophage population [108]. In this regard, the single-cell RNA sequencing analyses performed by Saleh et al. have provided a deeper map of skeletal muscle cellular diversity [114]. Using unsupervised clustering, the authors identified ten different macrophage subtypes. Among them, the M1-like and M2c-like clusters were mainly evident in dystrophic muscles, constituting approximately 20% and 17% of the macrophage population versus 4% and less than 1% in the healthy muscle. The M1-like cluster is characterized by the expression of *Ly6c2*, *Ccr2*, *Arg1*, and *Vcan* expression and low *Cx3cr1* expression, consistent with the results previously described by Juban et al. [108,114]. These findings highlight the complexity of macrophage subpopulations within dystrophic environments and emphasize the role of dynamic macrophage phenotypes in the pathological progression of muscular dystrophy [114].

It is interesting to highlight that recent work in mdx mice has shown that macrophages exhibit cardinal features of trained immunity [115]. The innate immune system has developed to recognize invading pathogens through a set of germline-encoded receptors called pattern recognition receptors (PRRs). These PRRs can detect structural motifs shared by numerous microorganisms [116]. It is becoming increasingly evident that PRRs not only detect and respond to exogenous pathogens but also recognize endogenous DAMPs released from injured tissues [117], resulting in an amplified immune response through increased production of pro-inflammatory cytokines. This process has been termed as “trained immunity”, a de facto innate immune memory [118]. The term “trained immunity” has been also recently proposed as another key factor that alters BMDMs, leading to transcriptional hyperresponsiveness associated with metabolic and epigenetic remodelling [115]. One of the major classes of PRRs is the Toll-like receptor (TLR) family. Among them, the best studied is probably Toll-like receptor 4 (TLR4) [116]. Regarding this, Bhattarai et al. have proposed that DAMPs, released from damaged muscles, are able to induce remodelling of the epigenetic, metabolic, and functional inflammatory profiles of BMDMs of mdx mice in a TLR4-dependent manner [115]. Seminal works have previously stated that macrophages can undergo epigenetic imprinting to confer trained immunity by regulating TLR4 [119,120,121,122]. In agreement with previously published data [108,114], BMDMs from mdx mice show an exaggerated response to external inflammatory stimuli, exhibiting significantly higher basal expression of both pro-inflammatory and anti-inflammatory genes compared to those from wild type mice [115]. These changes would take place centrally in the BM at a distance from the pathological muscle microenvironment, introducing a new layer of complexity. Before entering the affected muscle tissue, BMDMs in dystrophic mice undergo significant epigenetic reprogramming, which leads to substantial functional changes in these cells [115]. Collectively, these findings bolster the hypothesis that DAMP-mediated activation of the innate immune system via TLR4 signalling is a critical factor in the pathogenesis of DMD. Since mdx mice partially recapitulate DMD pathology in humans, further analysis in patients could provide additional information about the presence of trained immunity in DMD sufferers.

Finally, misregulation of fibrocytes activity has also been implicated in DMD progression [86]. This subset CD45^+^/CollagenI^+^ of macrophages appears to play a pathological role in maintaining chronic inflammation and driving progressive fibrosis due to the release of pro-inflammatory and pro-fibrotic cytokines in dystrophic diaphragm, thus promoting the ECM gene expression by fibroblasts [86].

## 5. Macrophages as Therapeutic Targets for Dystrophy DMD Treatment

Currently, one of the main therapeutic approaches in DMD is focused on mitigating chronic inflammation and fibrosis by using glucocorticoids, the only treatment that has shown a delay in disease progression, being able to prolong the ambulatory capacity of patients and modestly improve their cardiopulmonary function [96,97,98,99,123,124]. It has been proposed that glucocorticoids promote a shift from M1 to M2 anti-inflammatory macrophages, as observed in patients treated with prednisone (0.75 mg/kg/day) for 6 months [125]. Unfortunately, these potent anti-inflammatory drugs have significant side effects such as adrenal suppression, growth impairment, poor bone health, and metabolic syndrome [123,126,127,128,129]. The adverse effects of glucocorticoids on bone formation and atrophic effects on skeletal muscle are thought to be mediated by increased levels of active GSK3β, which reduces the protein synthesis rate by inhibiting eukaryotic transcription factor 2B-dependent translation [130,131,132]. In addition, glucocorticoids disturb skeletal muscle metabolism and hamper myogenesis and muscle regeneration by stimulating the AKT1/FOXO1 pathway, which decreases protein synthesis and increases protein catabolism, thus being responsible for the seemingly contradictory muscle weakness and atrophy observed in patients treated with this drug [133]. Therefore, it is crucial to identify alternative medications that can modulate the inflammatory response while minimizing adverse side effects. In this scenario, therapeutic approaches that harness macrophages are beginning to emerge.

M1 macrophages significantly exacerbate muscle injury in the mdx mouse model. However, the presence of M2c macrophages, which appear during tissue repair, suggests that the factors governing the balance between M1 and M2c phenotypes might impact the disease severity. Research has demonstrated that during the early acute phase of muscle pathology in mdx muscle, there is an increase in M1 macrophages. These macrophages are chiefly responsible for the inflammatory damage inflicted on dystrophic muscle fibers, primarily due to nitric oxide production by iNOS [45,101].

IL-10 has been proposed as a potential therapeutic tool for DMD treatment, since it has been shown that IL-10 deactivates cytolytic M1 macrophages, suppresses the production of pro-inflammatory cytokines [134,135], and facilitates the switch to the M2c phenotype [136,137] (Table 1). Villalta et al. have demonstrated that IL-10 deactivates the M1 phenotype in mdx muscle macrophages, inhibits the expression of iNOS, and modulates muscle cell differentiation, thus reducing muscle damage at the early acute stage of the disease [101]. In a parallel work, the same laboratory proposed IFN-γ block as a way to enhance muscle regeneration during DMD and preserve muscle function [138] (Table 1). IFN-γ is a strong inducer of the M1 phenotype and is elevated in mice with mdx dystrophy. However, Villalta et al. demonstrated that, contrary to expectations, a null mutation in IFN-γ did not decrease the cytotoxicity of macrophages isolated from mdx muscle. Additionally, it did not reduce muscle fiber damage in vivo or enhance gross motor function during the early acute phase of pathology in mdx mice. However, an increased M2 phenotype activation during the regenerative stage of the disease was observed as well as an improvement on motor function at the later stage of the disease in IFN-γ-null mdx mice. Thus, the authors showed that IFN-γ is a strong negative regulator of macrophage activation to the M2 phenotype during the regenerative stage of mdx dystrophy [138].

TNF is another potent inflammatory cytokine that is elevated in the muscles of both DMD patients and mdx mice and appears to exacerbate damage to dystrophic muscle in vivo [139,140,141] (Table 1). Long-term treatment with the cV1q mouse-specific anti-TNF antibody in mdx mice subjected to extended exercise resulted in improved muscle function, as evidenced by enhanced exercise capacity. Additionally, there was a significant decrease in myofiber leakage, indicated by lower serum creatine kinase levels and a reduction in disease severity, characterized by decreased myofiber necrosis and fewer regeneration cycles. This was reflected in a larger proportion of unaffected myofibers observed at 90 days of age [142]. These data agree with previously reported experiments using the monoclonal human/mouse chimeric antibody infliximab (Remicade) or the soluble TNF-receptor Etanercept, which clearly delayed and greatly reduced the breakdown of dystrophic muscle in young dystrophic mdx mice [143,144] (Table 1).

In vivo pharmacological inhibition of TGF-β-activating enzymes in macrophages has also been shown to improve the dystrophic phenotype (Table 1). Several studies have demonstrated that suramin, an antiparasitic and antineoplastic drug that inhibits the ability of TGF-β1 to bind to its receptors, reduces fibrosis and promotes muscle regeneration following strain and laceration injuries [145,146,147,148,149,150]. In this sense, Taniguti et al. have also demonstrated that suramin attenuated fibrosis in the diaphragm and limb muscles of mdx mice and prevented exercise-induced functional muscle loss during later stages of the disease [151] (Table 1). More recently, Juban et al. revealed that an AMPK-LTBP4 axis in inflammatory macrophages controls the production of TGF-β1, which is further activated by and acts on fibroblastic cells, leading to fibrosis in DMD [108] (Table 1). These authors demonstrated that AMPK activation by metformin, an antidiabetic compound known to activate AMPK, or by its specific activator 991, strongly reduced *Ltbp4* expression and TGF-β1 production by BMDMs treated with mdx muscle homogenates [108,152,153]. In addition, LY6C^+^ macrophages isolated from these muscles exhibited a strong decrease in *Ltbp4* expression, which was associated with a reduction in their TGF-β1 production. As a result, there was a reduction in the expression of the pro-inflammatory marker TNF-α and an increase in the levels of the anti-inflammatory markers CD206 and CD301. Additionally, treated animals exhibited a decrease in necrotic and fibrotic regions within the muscle, alongside an increase in the cross-sectional area (CSA) of the regenerating myofibers [108](Table 1).

Recently Babaeijandaghi et al. have shown that a small-molecule CSF1R inhibitor, PLX73086, currently under clinical investigation to treat rare cancers, is able to ablate TIMD4^+^ and TIMD4^−^ muscle-resident macrophage cells [28] (Table 1). The authors demonstrated that administering CSF1R inhibitors to mdx mice shields dystrophic muscles from eccentric contraction-induced injury both ex vivo and in vivo. This protective effect likely results from altering the balance between damage-sensitive glycolytic fibers and damage-resistant glycolytic-oxidative fibers. The benefits of CSF1R inhibition were evident within a few months of treatment. Given that several short-term preclinical and clinical studies have shown that CSF1R inhibitors are well tolerated, these findings strongly suggest that CSF1R inhibitors could help preserve ambulatory function in patients with DMD [28,154,155,156].

On the other hand, Saclier et al. have demonstrated that the deletion of Nfix in macrophages of dystrophic mice delays the establishment of fibrosis and muscle wasting and increases grasp force [157,158] (Table 1). Previously, NFIX was identified as a link between RhoA-ROCK1-dependent phagocytosis and the macrophage phenotypic switch from the pro-inflammatory to anti-inflammatory phenotype [47]. In this sense, Saclier et al. demonstrated that *Nfix* expression by macrophages during the development of dystrophy is associated with its progression by acting on both myogenic cells and FAPs (Saclier et al., 2022) (Table 1). These findings are in agreement with results published by Rossi et al. that had previously shown how transgenic mice that overexpress NFIX exacerbate the dystrophic disease in vivo and how transgenic mice that present a lack of NFIX showed a milder phenotype compared to mdx controls [159] (Table 1). In addition, these authors also silenced *Nfix* in the tibialis anterior of dystrophic mice, inducing a striking rescue of dystrophic muscle morphology in terms of reduced infiltrates, centrally nucleated myofibers, and CSA distribution [159]. All together, these results showed that targeting Nfix in macrophages could represent a valid approach to delay muscle wasting and fibrosis in DMD.

The inflammatory response is mediated, to a large extent, by the activation of the NF-kβ pro-inflammatory pathway [160,161]. Interestingly, NF-κβ is activated among the earliest histological features of DMD in neonates, even years before the symptoms appear. This finding indicates that early intervention with NF-κB suppressive therapy could potentially prevent or postpone the onset of muscle dysfunction by enhancing muscle regeneration and reducing fibrosis [162]. In this regard, it is interesting to stress that pharmacological GSK3β inhibition was able to avoid gene transcription mediated by NF-kβ [163,164], thus inhibiting the production of pro-inflammatory molecules such as IL-1β, IL-6, TNF, and IL-12 as well as stimulating the production of anti-inflammatory cytokines such as IL-10 in stimulated monocytes [165,166]. In agreement with these results, our group has recently shown that treatment with isolecanoric acid (ILA), a natural GSK3β inhibitor isolated from a fungal extract, decreases the expression of inflammatory cytokines TNF-α, IL-1β, and MCP-1 in dystrophic macrophages and diminishes muscle fibrosis in dystrophic mice [167,168] (Table 1). The notion that NF-kβ pathway disruption reports numerous benefits in dystrophic muscle is also reinforced by previous studies that showed how the inhibition of this pro-inflammatory pathway produces beneficial effects on functional, biochemical, and morphological parameters and is associated with a lower macrophage number and a better phenotype of the muscle in mdx mice [109,140]. Interestingly, our work also showed that ILA treatment boosts muscle regeneration in mdx dystrophic mice via the Wnt/β-catenin pathway. Given that DMD is a complex condition characterized by muscle fiber fragility, a harmful inflammatory environment, and diminished myogenic capacity of MuSCs, the administration of ILA is proposed as a potentially effective therapeutic strategy for DMD [167].
ijms-25-10393-t001_Table 1Table 1Molecular targets whose modulation ameliorates the dystrophic phenotype via macrophage activity.Molecular TargetStrategy/EffectReferencesIL-10Treatment suppresses the production of pro-inflammatory cytokines.Murray 2006; Mosser, Zhang 2008 [134,135]Treatment facilitates the switch to the M2c phenotype.Mantovani et al., 2004; Lang et al., 2002 [136,137]Treatment inhibits the expression of iNOS, thus enhancing muscle regeneration during DMD and preserving muscle function.Villalta, Rinaldi, et al., 2011 Villalta, Deng, et al., 2011 [101,138]IFN-γNull mutation increases M2 phenotype activation during the regenerative stage in mdx [138].Villalta, Deng, et al., 2011 [138]TNFcV1q anti-TNF antibody improves muscle function, reduces myofiber leakiness, and reduces DMD severity.Radley, et al., 2008 [142]Infliximab anti-TNF antibody and/or Etanercept soluble TNF-receptor delays and reduces the breakdown of dystrophic muscle in young dystrophic mdx mice.Grounds et al., 2004; Hodgetts et al., 2006 [143,144]TGF-βSuramin inhibits the ability of TGF-β1 to bind to its receptors and promotes muscle regeneration, attenuates fibrosis in the diaphragm and limb muscles, and prevents exercise-induced functional muscle loss of mdx mice.Chan et al., 2003; 2005; Nozaki et al., 2008; Coffey et al., 1987; La Rocca et al., 1990; McGeary et al., 2008;Taniguti et al., 2011 [145,146,147,148,149,150,151]Metformin and/or activator 991 decreases TNF-α production, increases CD206 and CD30 production, decreases the necrotic and fibrotic areas, and increases cross-sectional area in mdx mice.Foretz et al., 2014; Guigas et al., 2016; Juban et al., 2018 [108,152,153]CSF1RPLX73086 ablates TIMD4^+^ and TIMD4^−^ muscle-resident macrophages cells, thus shielding dystrophic muscles from eccentric contraction-induced injury.Babaeijandaghi et al., 2022 [28]NFIXDeletion of Nfix in macrophages of dystrophic mice delays the establishment of fibrosis and muscle wasting and increases grasp force.Saclier et al., 2022; Gronostajski 2000 [157,158]Nfix silencing rescues dystrophic muscle morphology in terms of reduced infiltrates, centrally nucleated myofibers, and CSA distribution.Rossi et al., 2017 [159]GSK3βILA decreases the expression of pro-inflammatory cytokines TNF-α, IL-1β, and MCP-1, diminishes muscle fibrosis, and boosts muscle regeneration in mdx.Matias-Valiente et al., 2023 [167]TLR4TLR4 ablation and/or Glycyrrhizin treatment decreases macrophage accumulation in mdx mice, also causing macrophages to acquire an anti-inflammatory phenotype.Giordano et al., 2015 [112]


We have previously discussed how trained immunity induces remodelling of the epigenetic, metabolic, and functional inflammatory profiles of BMDMs of mdx mice in a TLR4-dependent manner [115]. Interestingly, years before, Giordano et al. proposed targeting either TLR4 receptor or its endogenous ligands as a new therapeutic strategy to slow DMD progression [112] (Table 1). The authors showed that dystrophin-deficient mdx mice exhibited elevated TLR4 expression in their muscles. Genetic deletion of TLR4 led to reduced inflammation and improved various facets of the dystrophic disease. Specifically, TLR4 knockout in mdx mice resulted in a significant decrease in macrophage accumulation within the dystrophic muscles and an altered macrophage activation profile, marked by increased CD206 expression, which is associated with an anti-inflammatory phenotype. This change in macrophage behaviour was accompanied by favourable outcomes, such as reduced fibrogenesis and enhanced muscle force generation [112]. They also demonstrated that dystrophic muscles have increased levels of the endogenous TLR4 ligand HMGB1, a non-histone nuclear factor secreted by monocytes via a non-classical, vesicle-mediated secretory pathway that can behave as a pro-inflammatory cytokine [169,170]. Remarkably, administering Glycyrrhizin, a pharmacological inhibitor of HMGB1, to mdx mice yielded benefits comparable to those observed with TLR4 ablation, but only when TLR4 function was intact. These findings collectively reinforce the hypothesis that DAMP-driven activation of innate immunity via TLR4 signalling significantly contributes to the pathogenesis of DMD [112] (Table 1). In this regard, it is important to stress Bhattarai et al. have recently shown that BMDMs of mdx maintain a heightened response to external inflammatory stimuli even when they are removed from their in vivo environment and transplanted in wild type animals. This has very important implications for the therapeutic goal of restoring dystrophin expression in patients, because even if the muscle fibres could produce dystrophin again, the patients’ macrophages would continue to have a super-reactive phenotype that would still hinder their recovery.

## 6. Concluding Remarks and Future Research

Although our knowledge of the origins, functions, and activation mechanisms of skeletal muscle macrophages is still limited, we certainly know that these cells play diverse and essential roles in muscle regeneration, contributing to inflammation, phagocytosis, stem cell activation, and tissue repair processes. The dynamic nature of macrophage responses during muscle regeneration highlights their importance in orchestrating complex cellular interactions required for effective muscle repair. Therefore, understanding the balance between pro-inflammatory and anti-inflammatory macrophage phenotypes is crucial for optimizing muscle regeneration outcomes. More efforts are needed to decipher how this balance is deregulated in disease states such as DMD and identify potential interventions to modulate macrophage behaviour for therapeutic benefit. Inflammatory monocytes and resident tissue macrophages are key regulators of fibrosis. Fibrosis in DMD is a multifaceted process influenced by mechanical, humoral, and cellular factors that begins with injured myofibers and is marked by muscle cell necrosis and inflammation. In addition, the regenerative potential of the muscle in DMD is hampered by the limited number and capacity of MuSCs. Future research employing advanced technologies like lineage tracing, single-cell high-throughput analysis, and spatial transcriptomics is expected to enhance our understanding of the intricate interplay between macrophages and muscle regeneration, paving the way for innovative therapeutic strategies to improve outcomes for individuals with muscle disorders.

## Figures and Tables

**Figure 1 ijms-25-10393-f001:**
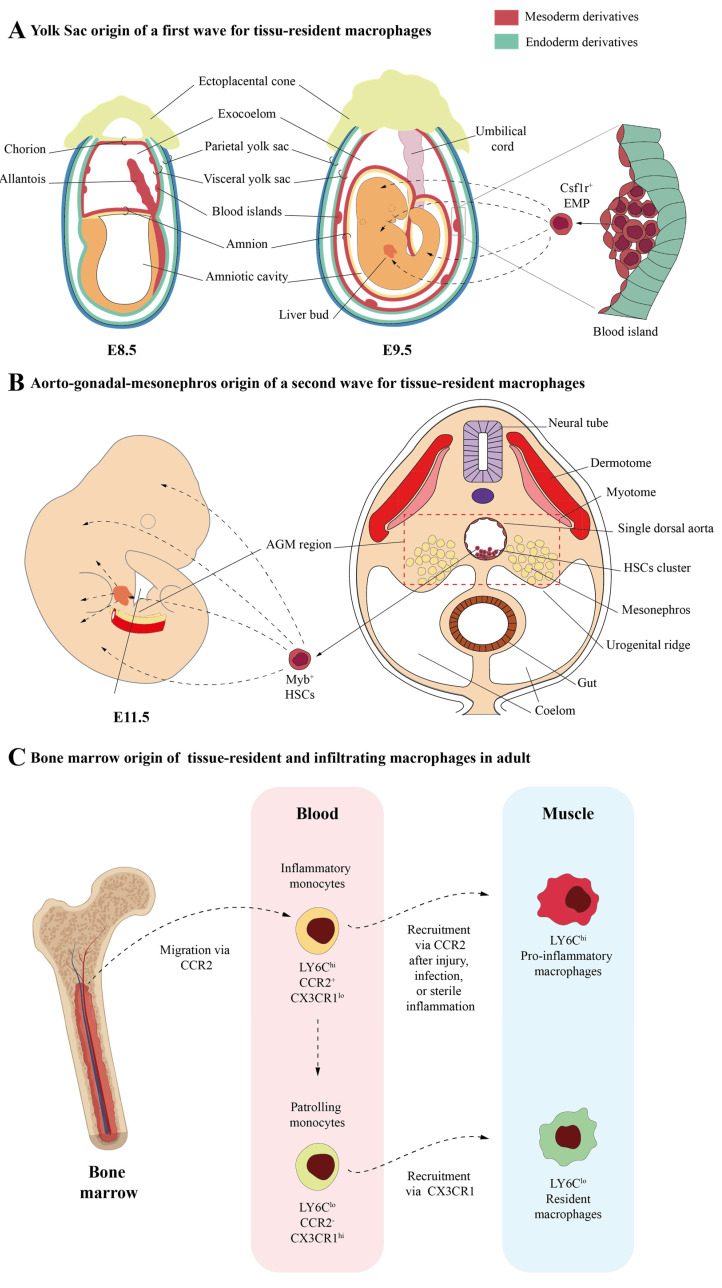
Embryonic and adult origins of macrophages. Tissue-resident macrophages arose from several origins, resulting in diverse macrophage populations (well reviewed by Kierdorf K. et al. and Wynn TA et al. [1,2]). (**A**) Initial precursors of tissue-resident macrophages emerge from Tie2^+^ hemogenic/endothelial progenitors located in the blood islands of the YS as early E7.5 in mice. Soon after, these Tie2^+^ progenitors give rise to Myb-independent EMPs, which are detected at E8.5 in YS and identified by Csf1r expression. This EMP migrates to the embryo proper and seeds its tissues from E9.5 onwards [1,2]. (**B**) Other resident macrophages arise from a second wave of hematopoietic stem cells (HSCs) that emerge in a Myb-dependent manner at E10.5 dpc in the AGM region. This myeloid progeny can be detected between E10.5 and E14.5. (**C**) BMDMs generated from peripheral blood monocytes in adult stages are also found in adult tissues. Blood monocytes consist of two principal subsets, LY6C^hi^/CCR2^+^/CX3CR1^lo^ and LY6C^lo^/CCR2^−^/CX3CR1^hi^ [19,20]. In the steady state, LY6C^hi^/CCR2^+^/CX3CR1^lo^ leave the BM in a CCR2-dependent manner. These inflammatory monocytes, in response to injury, infection, or sterile inflammation, quickly enter in tissues and differentiate into infiltrating-activated macrophages [19,20,21]. LY6C^hi^/CCR2^+^/CX3CR1^lo^ cells can also differentiate into circulating LY6C^lo^/CCR2^−^/CX3CR1^hi^ monocytes in blood [22,23]. These cells are patrolling monocytes that can be recruited into normal tissues by the interaction of the complementary pair CX3CR1/CCL3 in an LAF/ICAM1-dependent manner, becoming resident macrophages [19,20,24]. Human macrophages that do not express LY6C and CD14^hi^/CD16^low^ or CD14^low^/CD16^hi^ subsets of monocytes can be considered as LY6C^hi^ and LY6C^lo^ equivalent subsets, respectively [25]. AGM: aorto–gonadal–mesonephros; BM: bone marrow; BMDMs: bone marrow (monocyte)-derived macrophages; EMPs: erythromyeloid progenitors; YS: yolk sac.

**Figure 2 ijms-25-10393-f002:**
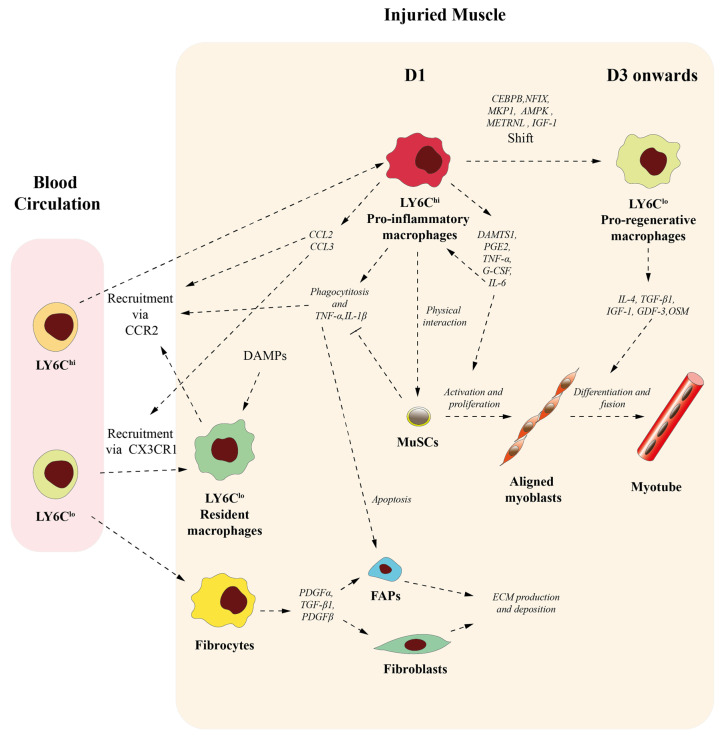
Circulating monocytes, tissue-macrophages, and muscle cells interplay during muscle regeneration. At day 1 (D1) after muscle damage, LY6C^hi^ circulating monocytes enter into muscle tissue, where they become LY6C^hi^ pro-inflammatory macrophages that (1) phagocyte muscle debris, (2) releasing a cocktail of cytokines which increase monocyte recruitment, promote MuCS activation/proliferation, and inhibit FAP apoptosis. In addition, in response to DAMPs, LY6C^lo^ resident macrophages contribute to enhance monocyte recruitment. Furthermore, fibrocytes derived from LY6C^lo^ release cytokines that promote FAPs and fibroblast ECM deposition. After day 3 (D3), LY6C^hi^ pro-inflammatory macrophages shift to LY6C^lo^ pro-regenerative macrophages, inducing myoblast differentiation. DAMPs: damage-associated molecular patterns; ECM: extracellular matrix; FAPs: fibro/adipogenic progenitors; MuSCs: muscle stem cells.

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
