# Peer review of "Macrophages in the Context of Muscle Regeneration and Duchenne Muscular Dystrophy"

_ijms, 2024, doi:10.3390/ijms251910393_

Round 1
Reviewer 1 Report
Comments and Suggestions for Authors
The manuscript provides a review of macrophages' role in muscle repair and their involvement in Duchenne Muscular Dystrophy (DMD). The topic is timely and relevant, as understanding the dynamic roles of macrophages could pave the way for novel therapeutic strategies for DMD as described. The manuscript is well-organized and covers a broad spectrum of topics related to macrophage origin, plasticity, muscle repair, and the pathophysiology of DMD.
However, there are several areas where the manuscript could be improved.
Comments:
1-The manuscript could benefit from additional figures and tables to visually represent the concepts discussed, particularly a figure for paragraph "3.2.-. Macrophages participate in muscle stem cell activation and differentiation" to enhance the reader’s understanding of the complex interactions between macrophages and muscle cells as well as a table summarizing the paragraph "5.-. Macrophages as therapeutic targets for dystrophy DMD treatment".
2-The authors should not only summarize current knowledge but also critically evaluate the limitations of existing studies and suggest potential directions for future research.
3-Adding references such as Borrego et al. (2022) could provide recent critical insights into how different macrophage phenotypes, influenced by the extracellular environment, can promote or inhibit myogenic cell proliferation and differentiation. and are actively involved in the regenerative process by impacting myogenic cell behavior through changes in their secretome.
Borrego, I., Frobert, A., Ajalbert, G., Valentin, J., Kaltenrieder, C., Fellay, B., Stumpe, M., Cook, S., Dengjel, J., & Giraud, M. N. (2022). Fibrin, Bone Marrow Cells, and Macrophages Interactively Modulate Cardiomyoblast Fate. Biomedicines, 10(3), 527. https://doi.org/10.3390/biomedicines10030527
Comments on the Quality of English Language
NA
some minor typo such as expressión
Author Response
Comment 1: The manuscript could benefit from additional figures and tables to visually represent the concepts discussed, particularly a figure for paragraph "3.2.-. Macrophages participate in muscle stem cell activation and differentiation" to enhance the reader’s understanding of the complex interactions between macrophages and muscle cells as well as a table summarizing the paragraph "5.-. Macrophages as therapeutic targets for dystrophy DMD treatment".
Response 1: Thank you for your frank and constructive comments. We apologize for the lack of clarity in this issue. The paragraph 3.2, as well as the paragraphs 3.1 and 3.3, are supported by the attached figure 2. Nonetheless, by reading the text again, we have come to realize that we did not clearly indicate this fact in the manuscript. This inevitably leads to misunderstandings. We apologize for the inconvenience. We have added the term “Figure 2” in the text in those sections that require graphic support to facilitate reading. The changes are highlighted in red. In addition, we have summarized the paragraph "5.-. Macrophages as therapeutic targets for dystrophy DMD treatment" with a table (Table 1) in the new version of the manuscript as the reviewer indicates.
Comment 2: The authors should not only summarize current knowledge but also critically evaluate the limitations of existing studies and suggest potential directions for future research.
Response 2: In accordance with reviewer´s suggestion, additional sentences evaluating the limitations of existing studies and suggesting potential directions for the future have been included in the new version of the manuscript in lines 157-162 (page 6), lines 334-337 (page 11), and lines 437-439 (page 15)
Comment 3: Adding references such as Borrego et al. (2022) could provide recent critical insights into how different macrophage phenotypes, influenced by the extracellular environment, can promote or inhibit myogenic cell proliferation and differentiation. and are actively involved in the regenerative process by impacting myogenic cell behavior through changes in their secretome.
Borrego, I., Frobert, A., Ajalbert, G., Valentin, J., Kaltenrieder, C., Fellay, B., Stumpe, M., Cook, S., Dengjel, J., & Giraud, M. N. (2022). Fibrin, Bone Marrow Cells, and Macrophages Interactively Modulate Cardiomyoblast Fate. Biomedicines, 10(3), 527. https://doi.org/10.3390/biomedicines10030527
Response 3: We thank the reviewer for his/her recommendation. The inclusion of the work cited by him/her undoubtedly substantially improves the quality of our paper. We have inserted new text describing the most important aspects of the proposed paper and the corresponding citation (pages 8-9 lines 236-248) and References (number 54) section in the new version of the manuscript
Reviewer 2 Report
Comments and Suggestions for Authors
Dear Authors,
It is with great interest that I read your work.
Please find here some suggestions:
1) Please write out all abbreviations
2) Please foresee a list with abbreviations
3) Line 237: Please replace X by the correct figure number
4) Please correct all the extra spaces between words
5) Line 313: Could it be possible to specify "of what" at the end of the sentence?
6) Line 343: Could it be possible to rephrase please?
Best wishes,
Reviewer
Comments on the Quality of English LanguageSome phrases should be corrected.
Author Response
Comment 1: 1) Please write out all abbreviations.
Response 1: Thank you for your frank and constructive comments. In accordance with the reviewer´s recommendation we have written out all abbreviations throughout the manuscript. The changes are highlighted in red.
Comment 2: 2) Please foresee a list with abbreviations
Response 2: In agreement with the reviewer’s comment, we have added a list with abbreviations on the first page of the manuscript.
Comment 3: 3) Line 237: Please replace X by the correct figure number
Response 3: We apologize for this mistake. In the revised manuscript, we have replaced figure X by figure 2 in the line 226 of the new version of the manuscript
Comment 4: 4) Please correct all the extra spaces between words
Response 4 : We apologize for this inaccuracy. In the revised manuscript, we have corrected all the extra spaces between the words.
Comment 5: 5) Line 313: Could it be possible to specify "of what" at the end of the sentence?
Response 5: We thank the reviewer for pointing this out. Indeed, the word ‘of’ at the end of the sentence is not necessary and must be left out. We apologize for the mistake. We have erased this word y the new version of the manuscript. Line 313
Comment 6: 6) Line 343: Could it be possible to rephrase please?
Response 6: As the reviewer suggest, we have rephrased the sentence. In the new version of the manuscript, it reads “The dystrophic phenotype is further exacerbated by impaired muscle regeneration, since dystrophin is essential for the asymmetric division of MuSCs and the generation of committed myogenic progenitors”. Lines344-346
Reviewer 3 Report
Comments and Suggestions for Authors
This revision is presenting an actual view of the dynamic functions of macrophages in muscle regeneration and their implications in the dystrophic process. It is well known the important role of macrophages in muscle regeneration by orchestrating inflammation, phagocytosis, and tissue repair processes. Also, the capacity of these cells to switching from pro-inflammatory (M1) to anti-inflammatory (M2) states during muscle repair, influencing myoblast proliferation, differentiation, and myofiber formation. Understanding the complex interaction between macrophages and muscle cells is essential for developing targeted therapies for muscle disorders. Therefore, this revision if very important.
In addition to the well knows data on M1/M2 origin and function, new data from 2020 are presented showing the existence of a skeletal muscle-resident macrophage lineage with specific functionality. A more recent study using single cell-RNA seq analyses (2022), adds evidences that the skeletal muscle macrophages are divided in four subgroups, and not only the two known M1-like and M2-like groups.
The important role of the macrophage in muscle injury is well discussed. A proper inflammatory response, driven primarily by macrophage infiltration is crucial for the repair of acute skeletal muscle injuries. Without macrophage function, muscle regeneration is severely compromised, leading to significant muscle fibrosis. But resident macrophages are unable to fully compensate for pro-regenerative function, and infiltrating monocyte-derived macrophages are important for skeletal muscle regeneration after acute injury.
The role of macrophages on skeletal muscle homeostasis and muscle repair is well discussed, including Macrophaes contribution to the initiation and resolution of the inflammation, Macrophage participation in muscle stem cells activation and differentiation and the role of macrophages conditioning muscle repair-microenvironment.
The role of macrophages in DMD microenvironment is also well discussed. In Duchenne Muscular Dystrophy, muscle injuries occur asynchronously with the absence of effective regeneration, chronic inflammatory status and fibrosis. The pattern of macrophage phenotype switching from M1-like to M2like phenotype is more complex in the dystrophic muscle.
And finally, macrophages as therapeutic targets for dystrophy treatment is also well discussed.
This revision is very good and well written.
Author Response
Comment 1: This revision is presenting an actual view of the dynamic functions of macrophages in muscle regeneration and their implications in the dystrophic process. It is well known the important role of macrophages in muscle regeneration by orchestrating inflammation, phagocytosis, and tissue repair processes. Also, the capacity of these cells to switching from pro-inflammatory (M1) to anti-inflammatory (M2) states during muscle repair, influencing myoblast proliferation, differentiation, and myofiber formation. Understanding the complex interaction between macrophages and muscle cells is essential for developing targeted therapies for muscle disorders. Therefore, this revision if very important.
In addition to the well knows data on M1/M2 origin and function, new data from 2020 are presented showing the existence of a skeletal muscle-resident macrophage lineage with specific functionality. A more recent study using single cell-RNA seq analyses (2022), adds evidences that the skeletal muscle macrophages are divided in four subgroups, and not only the two known M1-like and M2-like groups.
The important role of the macrophage in muscle injury is well discussed. A proper inflammatory response, driven primarily by macrophage infiltration is crucial for the repair of acute skeletal muscle injuries. Without macrophage function, muscle regeneration is severely compromised, leading to significant muscle fibrosis. But resident macrophages are unable to fully compensate for pro-regenerative function, and infiltrating monocyte-derived macrophages are important for skeletal muscle regeneration after acute injury.
The role of macrophages on skeletal muscle homeostasis and muscle repair is well discussed, including Macrophaes contribution to the initiation and resolution of the inflammation, Macrophage participation in muscle stem cells activation and differentiation and the role of macrophages conditioning muscle repair-microenvironment.
The role of macrophages in DMD microenvironment is also well discussed. In Duchenne Muscular Dystrophy, muscle injuries occur asynchronously with the absence of effective regeneration, chronic inflammatory status and fibrosis. The pattern of macrophage phenotype switching from M1-like to M2like phenotype is more complex in the dystrophic muscle.
And finally, macrophages as therapeutic targets for dystrophy treatment is also well discussed.
This revision is very good and well written.
Response 6: We greatly appreciate the reviewer’s comments and observations and are pleased that he/she found our review of interest and suitable for publication.